# Phylogenetic-Related Divergence in Perceiving Suitable Host Plants among Five Spider Mites Species (Acari: Tetranychidae)

**DOI:** 10.3390/insects13080705

**Published:** 2022-08-05

**Authors:** Qi-Qi Hu, Xin-Yue Yu, Xiao-Feng Xue, Xiao-Yue Hong, Jian-Ping Zhang, Jing-Tao Sun

**Affiliations:** 1Department of Entomology, Nanjing Agricultural University, Nanjing 210095, China; 2College of Agriculture, Shihezi University, Shihezi 832003, China

**Keywords:** Y-tube olfactometer, two-choice disc, feeding preference, chemosensing system

## Abstract

**Simple Summary:**

Many spider mites are important agricultural pests in both fields and greenhouses worldwide and are diversified in their host plant range. How spider mites perceive their suitable host plants remains not completely clear. In this study, we found that spider mites cannot locate suitable host plants by volatile odours from a long distance, but they can use olfactory sensation in combination with gustatory sensation to make a precise selection for suitable host plants at a short distance. Highly polyphagous species showed strong sensitivity in sensing suitable host plants rather than the expected lowered sensitivity. We also found that the similarity among the five spider mite species in their performance in perceiving suitable host plants was highly correlated with their relative phylogenetic relationship.

**Abstract:**

Spider mites belonging to the genus *Tetranychus* infest many important agricultural crops in both fields and greenhouses worldwide and are diversified in their host plant range. How spider mites perceive their suitable host plants remains not completely clear. Here, through two-host-choice designs (bean vs. tomato, and bean vs. eggplant), we tested the efficacies of the olfactory and gustatory systems of five spider mite species (*T. urticae*, *T. truncatus*, *T. pueraricola*, *T. piercei*, and *T. evansi*), which differ in host plant range in sensing their suitable host plant, by Y-tube olfactometer and two-choice disc experiments. We found that spider mites cannot locate their suitable host plants by volatile odours from a long distance, but they can use olfactory sensation in combination with gustatory sensation to select suitable host plants at a short distance. Highly polyphagous species displayed strong sensitivity in sensing suitable host plants rather than the lowered sensitivity we expected. Intriguingly, our principal component analyses (PCAs) showed that the similarity among five spider mite species in the performance of perceiving suitable host plants was highly correlated with their relative phylogenetic relationships, suggesting a close relationship between the chemosensing system and the speciation of spider mites. Our results highlight the necessity of further work on the chemosensing system in relation to host plant range and speciation of spider mites.

## 1. Introduction

Spider mites belonging to the genus *Tetranychus* comprise many important agricultural pests, damaging a wide range of crops and ornamental plants. For instance, the two-spotted spider mite *Tetranychus urticae* has been considered one of the most destructive agricultural pests in the world [1] and can feed on more than 1150 plant species from 140 different plant families [2]. Additionally, several other species also commonly occur in the field, such as *Tetranychus kanzawai*, *Tetranychus truncatus*, *Tetranychus pueraricola,* and *Tetranychus evansi* [3,4,5,6,7].

Currently, 154 species are documented in the *Tetranychus* genus [8]. They exhibit high diversity in terms of host plant range, from a near specialist (*Tetranychus lintearius*) to an extreme generalist (*T. urticae*) [8], providing a suitable model to investigate the evolution of diet breadth and the underlying mechanisms. How do spider mites sense their suitable host plants? Studies regarding this fundamental question are scarce. Unlike insects, which commonly use antennae to perceive suitable plants by their volatiles, spider mites lack antennae. Gotoh et al. [9] studied the abilities of two host races of *T. urticae* (tomato and cucumber) to discriminate their respective hosts via a Y-tube olfactometer and two-choice disc experiments. They found that the two races can significantly perceive their respective favourite hosts in the two-choice disc experiment. In contrast, only the tomato host race displayed a weak preference for tomato in the Y-tube olfactometer experiment. Therefore, they considered that *T. urticae* could not perceive plants by plant volatiles alone but by both volatile and contact cues. As *T. urticae* is an extreme generalist, it might be unnecessary to discriminate against its favourite host plant when facing different host plants. Alternatively, the low genetic divergence between the two host races may not render a significant divergent preference for favourite plants. Furthermore, it remains unclear how other *Tetranychus* species (e.g., *T. evansi*) with a relatively narrower host range than *T. urticae* perceive suitable hosts.

In China, eight *Tetranychus* species commonly occur in the field, according to our previous extensive investigation, which covered 317 populations from 58 host plants (most plants are crops) during 2008–2017 [3]. Among them, *T. truncatus* was the predominant species, with an incidence of 45.7%, followed by *T. pueraricola* (21.4%), *T. kanzawai* (12.5%) and *T. urticae* (including both green and red, 4.5%) [3]. The remains include *Tetranychus ludeni*, *Tetranychus macfarlanei*, and *Tetranychus piercei*. The first three dominant species, *T. truncatus*, *T. pueraricola* and *T. kanzawai*, were much richer in the host plant range than the remaining species in China. The three species were found on 30 (for *T. truncatus*), 24 (for *T. pueraricola*) and 18 (for *T. kanzawai*) out of the 58 host plant species we investigated. Despite the extreme polyphagous nature of *T. urticae*, it was only found on seven host plant species for the green form and on 13 species for the red form, which is probably due to its invasive status in China and the loss of genetic variation after invasion [10]. *Tetranychus piercei* is a tropical and warm subtropical species of Southeast Asia and the Indonesian region, having a much narrower host range and mainly occurring on woody plants. We only found this species on eight out of the 58 plant species during our investigation [3]. It is an important species of banana [11] in China. Recently, we found a new invader, *T. evansi*, emerging as a new threat to solanaceous crops [12]. This species is only found on tomatoes in our recent investigation (data not published) and was widely considered a solanaceous specialist [13,14].

In the laboratory, *T. urticae*, *T. pueraricola*, *T. truncatus*, and *T. piercei* are more easily expanded on bean plants than on eggplants or tomatoes, indicating that bean plants are a more suitable host than the two others for these four species. In contrast, *T. evansi* has the highest fitness in tomatoes. Here, we ask the following questions: (1) How do spider mites perceive their suitable host plants by smell or taste? (2) Does the selectivity vary among species with different ranges and preferences of host plants? Specifically, whether highly polyphagous species displays lowered sensitivity in sensing suitable host plants than those with narrow host ranges? To answer these questions, we compared the host selectivity of five *Tetranychus* species with different ranges and preferences in their host plants using a Y-tube olfactometer and two-choice disc bioassays.

## 2. Materials and Methods

### 2.1. Experimental Plants

Tomato (*Solanum lycopersicum*) and eggplant (*Solanum melongena*) were grown individually in plastic pots 10 cm in diameter. Plants 10–30 cm in height (before the flowering stage) were used for the experiments. Kidney bean plants (*Phaseolus vulgaris*) were grown in plastic pots (30 × 30 × 20 cm). Kidney bean plants that were three to four weeks old (before the flowering stage) were used for spider mite rearing and experiments. All plants were grown in a greenhouse provided by Nanjing Agricultural University, with a long day photoperiod (16 L:8 D), temperature (25 ± 1) °C, and relative humidity (60 ± 10)%.

### 2.2. Spider Mites

*T**etranychus urticae* (green form), *T. truncatus*, *T*. *pueraricola* and *T*. *piercei* were reared on kidney bean plants placed in cages. *T. evansi* was reared on tomato plants in a cage. These spider mites were identified through a molecular approach. Briefly, we amplified and sequenced the nuclear ribosomal transcribed spacer (ITS) region according to Ge et al. [15] and then blasted these sequences against NCBI “nr” nucleotide sequence database. Sampling information is shown in Table 1. All spider mites were maintained in a greenhouse, which is provided by Nanjing Agricultural University, with long day photoperiod (16 L:8 D), temperature (25 ± 1) °C, and relative humidity (60 ± 10)%.

### 2.3. Y-Tube Olfactometer Experiments

We performed the olfactory selection test following Gotoh et al. [9] with slight modification under conditions of 25± °C, 30 ± 10% R.H. (Figure 1). We adopted a glass Y-tube olfactometer with a 0.5 cm uniform diameter, 5 cm trunk length, 5 cm branch length, and 75° branch angle. The two arms were connected to the bottle, and the leaves of fresh host plants with the same growth condition and weight of 2.35 ± 0.15 g (the maximum limit that the bottle can accommodate without damaging the plant materials) were selected as the odour source. The incision part of the leaves was wrapped with moist absorbable cotton, and the outer part was wrapped with a layer of aluminium foil to avoid leaf wilting due to water loss. The air was purified by activated carbon and humidified with distilled water. The rate of airflow was adjusted to 300 mL/min, and the air was blown for 15 min before the test to flush the olfactometer. 

At the commencement of the experiment, the spider mites were individually positioned on the starting point 2 cm from the opening of the Y-tube. When the spider mites reached the point beyond 1/2 of an arm and remained for more than 15 s or reached the outlet, the “choice” of odour that they moved towards was recorded. A spider mite was recorded as having no selection response if it did not make a choice within 5 min. After every fifth bioassay, we exchanged the arm containing the exposed odour with an arm containing the control odour. Chambers were cleaned and rinsed thoroughly in water after each observation. Each spider mite was tested only once in the experiment, and a total of 40 effective spider mites were tested in each group. 

### 2.4. Two-Choice Disc Bioassay

The choice test for spider mites was performed in plastic Petri dishes (15 cm diameter). A 1.5 cm diameter circular piece of host plant leaf was cut and placed in a Petri dish, which was lined at the bottom with thick soaked absorbent cotton. Parafilm was cut into a rectangle with a length of 4 cm and a width of 1 cm. The two sides of the rectangle touched the leaves of the kidney bean leaf disc and the other host plants. Twenty adult spider mite females, which were starved for two hours before the experiment, were introduced at the centre of the parafilm. The experiment was repeated eight times for each treatment. The number of mites on each leaf disc was recorded at 2, 6, 12, 24, 36, 48 and 72 h. The Petri dishes were placed into a climate-controlled cabinet at 25 ± 1 °C and 16 L:8 D.

### 2.5. Data Analysis

We performed χ^2^ tests in IBM SPSS Statistics 22 to analyse mite preference behaviours for both the Y-tube olfactometer experiments and two-choice disc experiment data. Significances were determined at three α levels, namely 0.05, 0.01 and 0.001. All graphs were plotted using GraphPad Prism 8 software. To further decipher the similarities among the five spider mites in response to the three tested plant species, we performed principal component analyses (PCA) for the two-choice disc experiment data and the combination of two-choice disc experiment data (averaged over the eight replicates for each species) and Y-tube olfactometer experiment data. This analysis was conducted in R v.4.1.0.

## 3. Results

### 3.1. Performance in Y-Type Olfactometer Bioassay

Our χ^2^ tests indicated no significant preference when the five spider mites were offered the choices between bean and tomato or between bean and eggplant (Figure 2, *p* > 0.05). Although without significance, all species showed a trend of preference for odour from bean plants to that from tomato plants, except *T. piercei,* which displayed a weak preference for odour from tomatoes when offered the choice between beans and tomatoes. Even though *T. evansi* showed an apparent preference for Solanaceae in the field [16], it displayed no significant preference for odour from tomato or eggplant plants. In contrast, the species tended to prefer odour from bean plants to odour from tomato or eggplant plants. The results suggest that the five species of spider mites cannot accurately distinguish between the volatile odours of the three host plants tested based on olfactory action alone.

### 3.2. Performance in Two-Choice Disc Bioassays

The five species showed clear preferences for suitable plants in the two sets of two-choice disc bioassays. *Tetranychus evansi* showed a significant preference for the two Solanaceae species when offered choices between beans and tomatoes or beans and eggplants across all time points (Figure 3I,J). However, the remaining four species showed a significant preference for the beans, except for *T. truncatus* at 2 h and *T. piercei* at 72 h when offered the choice between beans and tomatoes (Figure 3C,G). Although the four species all preferred beans, they varied in the extent of preference. We noticed a pattern: young evolutionary species tend to be more sensitive to perceiving suitable plants.

To further visualise the differences intuitively among the five species in their performances in the two sets of two-choice disc bioassays, we also performed PCAs on the data of each set. The first two PCs consistently accounted for 91.86% of the total variations in both two-choice disc bioassays (Figure 4A,B). Consistent with our observation, the five species can be divided into two groups based on their performance in the two sets of two-choice disc bioassays, particularly in the choice between beans and eggplants. *Tetranyhus evansi* constituted one group that diverged from the other species because it preferred the two Solanaceae species. The remaining four species constituted the other group, with shallow divergence among each other. We noticed that *T. piercei* and *T. truncatus* were much closer to *T. evansi* than the other two species, as shown in Figure 4A, particularly in the choice between beans and tomatoes, likely due to their lower preference for beans than *T. urticae* and *T. pueraricola*.

### 3.3. Overall Performance

To explore the difference in overall performance among the five species, we also performed a PCA by combining the two-choice disc and Y olfactometer bioassays. The first two PCs accounted for 94.73% of the total variation (Figure 4C). *Tetranyhus evansi*, which was highly different from the other four species in selection behaviour due to the strong preference for the two solanaceous plants in the two-choice disc bioassays, is separated along PC1. *Tetranyhus urticae*, *T. pueraricola* and *T. truncatus*, which displayed high sensitivity in recognising suitable plants (bean) in the two-choice disc bioassays, were clustered together (Figure 4C,D), and *T. urticae* and *T. pueraricola* were the most proximate in the PCA plot. *Tetranyhus piercei* was much closer to the cluster consisting of *T. urticae, T. pueraricola,* and *T. truncatus* than to *T. evansi* but was separated from the three species along PC2.

## 4. Discussion

In phytophagous insects, chemosensory perception, including olfaction and gustation, plays a critical role in host-plant selection [20,21], in which olfaction is usually responsible for host orientation, and gustation is responsible for host selection [22,23,24]. Compared to insects, spider mites lack antennae and have more limited mobility, with potential repercussions on the evolution of chemosensing [25]. In support of this, our Y-type olfactometer bioassay results showed that the five spider mites could not orientate suitable host plants by volatile odours, indicating that olfactory sensation did not play a key role in sensing host plants from long distances by spider mites. However, its function of assisting gustatory sensation in sensing host plants cannot be ruled out.

Spider mites have neuron-rich setae harbouring cuticular pores on the palpa and legs [26] that are reminiscent of functionally characterised chemosensory sensilla present on insect appendages [27]. In our two-choice disc bioassays, we also observed that spider mites randomly wandered on the bridge between the two discs at the beginning and determined whether to climb on the leaf once they touched the leaf with their fore legs. This phenomenon probably indicates that spider mites can sense volatile odours over short distances using their forelegs. However, after they climbed on the leaf disc, considerable proportions of spider mites chose to leave. This also reflects that spider mites not only rely on olfactory sensation for initial host plant selection but also use gustatory sensation for more precise selection. This characteristic might have evolved in concert with the passive dispersal propensity of spider mites, in which spider mites primarily rely on wind and human activities over long distances [28,29].

We expected that more polyphagous species may have decreased sensitivity in perceiving suitable host plants, considering that they possess a strong ability to adapt to new host plants; thus, it is not necessary for them to recognise their suitable host plants. As opposed to our expectation, although extremely polyphagous, *T. urticae* perceives their suitable host plants most sensitively. This is probably linked to the massive number of gustatory receptors in the *T. urticae* genome [25]. A total of 689 gustatory receptors were identified, and the genomic distribution of *T. urticae* gustatory receptors indicates recurring bursts of lineage-specific proliferation [25]. As only one complete genome is available for the *Tetranychus* genus to date, more genomes will help dissect the relationships between gustatory receptors and feeding preferences from an evolutionary perspective.

The positions of the five spider mite species in the PCA plot were highly associated with their relative phylogenetic relationships previously inferred by nuclear and mitochondrial DNA sequences (Figure 4C,D) [17,18,19]. This result probably reflects a close relationship between the chemosensing systems and speciation of spider mites. A substantial proportion of speciation events in insect herbivores can be attributed to shifts towards novel host plant taxa [30]. Host races have also been widely identified in a myriad of generalist herbivores [31,32], including *T.urticae* [33]. However, the role of the chemosensing system in forming process receives little attention. Function alternation of the chemosensing system might promote divergence in host choice or preference in nature, thus decreasing the opportunity for co-occurrence on the same host plants, restricting gene flow between taxa with divergent chemosensing systems and promoting genetic differentiation. Theoretically, this is expected to be more pronounced in the formation of sympatric host races in generalist taxa. Therefore, the role of the chemosensing systems in host range variation and host plant-associated speciation of spider mites needs to be investigated in the future.

## 5. Conclusions

Collectively, we found that spider mites cannot locate suitable host plants based on volatile odours from long distances, but they can use olfactory sensation in combination with gustatory sensation to make a precise selection for suitable host plants over short distances. The similarity in performance in selecting suitable host plants among the five spider mite species was highly correlated with their relative phylogenetic relationships. Highly polyphagous species displayed strong sensitivity in sensing suitable host plants rather than lowered sensitivity as expected. Our results highlight the necessity of future work on the chemosensing system in relation to host plant range and speciation of spider mites.

## Figures and Tables

**Figure 1 insects-13-00705-f001:**
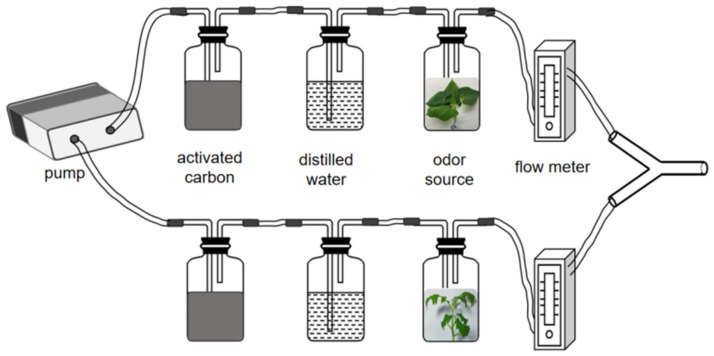
Schematic of the Y type olfactory device.

**Figure 2 insects-13-00705-f002:**
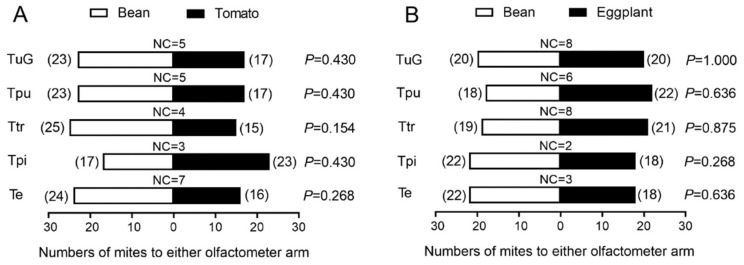
Olfactory selection of five species of spider mites (**A**) between bean and tomato and (**B**) between bean and eggplant plants. TuG, Tpu, Ttr, Tpi and Te represent *T. urticae* (green form), *T. pueraricola*, *T.*
*truncatus*, *T. piercei* and *T.*
*evansi*, respectively. The numbers of individuals selected for each plant are labelled in the brackets near the bars. NC, number of individuals that made no selection.

**Figure 3 insects-13-00705-f003:**
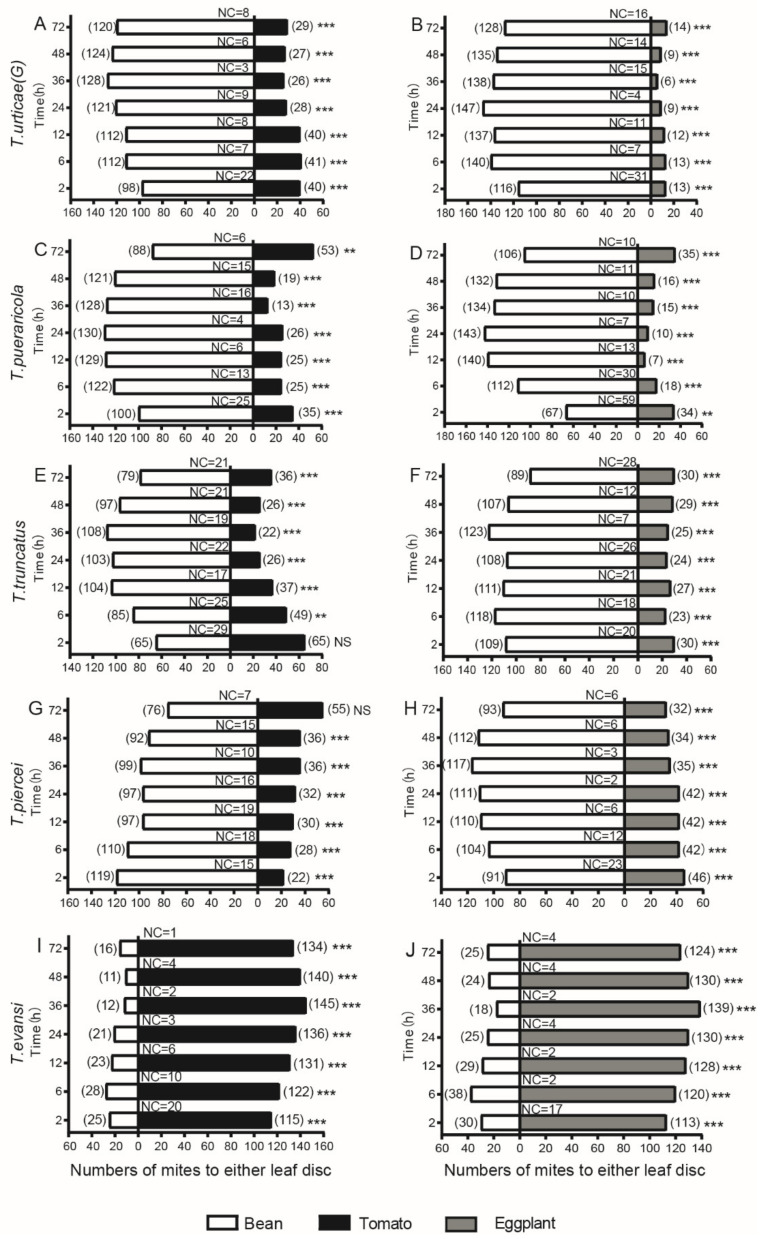
Selectivity of five species of spider mites to bean and tomato within 72 h. **, *** represent *p* < 0.01 and *p* < 0.001, respectively; NS, no significant difference; NC, number of individuals that made no selection. (**A**,**B**) for *T. urticae*; (**C**,**D**) for *T. pueraricola*; (**E**,**F**) for *T.*
*truncates*; (**G**,**H**) for *T. piercei*; (**I**,**J**) for *T.*
*evansi*.

**Figure 4 insects-13-00705-f004:**
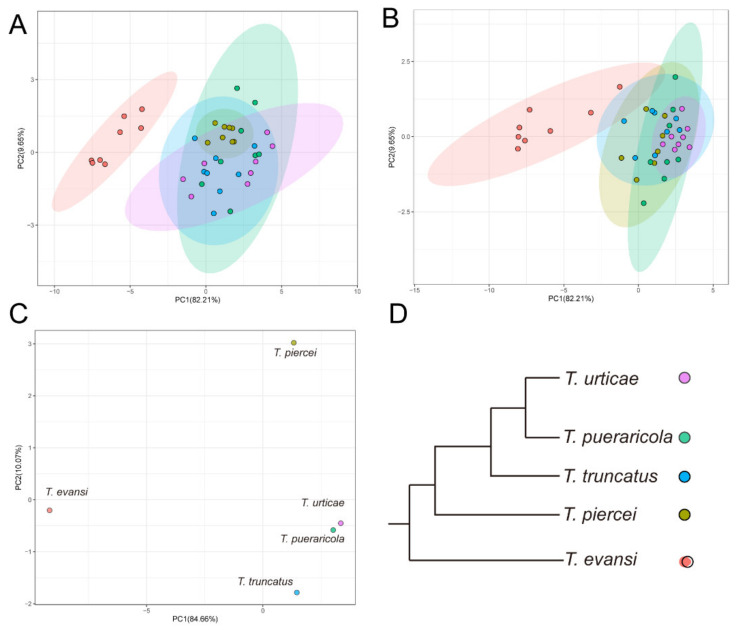
PCA results of host selection of spider mites. (**A**) PCA of leaf disc selection of bean and tomato by spider mites. (**B**) PCA of leaf disc selection of bean and eggplant by spider mites. (**C**) PCA of leaf disc selection and Y-tube olfactory selection. (**D**) Relative positions of five species of spider mites in phylogenetic trees inferred by DNA sequences. The tree was depicted according to the previous phylogenetic studies using 18S and 28S rRNA genes, mitochondrial cytochrome c oxidase subunit 1 gene [17], RNA-Seq data [18] and mitochondrial genome [19] with a maximum likelihood method.

**Table 1 insects-13-00705-t001:** Sampling information of spider mites used in this study.

Species	Sampling Date	Location	Host Plant	Coordinates
*T. urticae* (green form)	2015.08	Rizhao, Shandong	*Arachis hypogaea*	119.46° E, 35.42° N
*T. pueraricola*	2016.08	Huizhou, Guangdong	*Solanum melongena*	114.74° E, 22.98° N
*T. truncatus*	2016.08	Huizhou, Guangdong	*Vigna sinensis*	114.42° E, 23.12° N
*T. piercei*	2020.06	Sanya, Hainan	*Trachycarpus fortunei*	118.80° E, 32.06° N
*T. evansi*	2017.07	Yaan, Sichuan	*Lycopersicum esculentum*	103.00° E, 29.98° N

## Data Availability

The data that support the findings of this study are available from the corresponding author upon reasonable request.

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
