# Peer review of "Phylogenetic-Related Divergence in Perceiving Suitable Host Plants among Five Spider Mites Species (Acari: Tetranychidae)"

_insects, 2022, doi:10.3390/insects13080705_

Round 1
Reviewer 1 Report
Review of the MS: Phylogenetic related divergence in perceiving suitable host plants among five spider mites species.
The MS is about the mites of the genus Tetranychus which are important pests of many agricultural crops. The Authors tested how five spider mite species perceive their suitable host plant. They found that these species cannot locate their suitable host plants by volatile odours from a long distance, but can use olfactory sensation in combination with gustatory sensation to select suitable host plants at a short distance.
Title: In my opinion the title of the MS suggests that the Authors include their own phylogenetic study, while the Methods do not explain how the tree presented in Figure 4 was obtained. The title under this figure does not say what type of tree is this (for example neighbour joining tree or other?).
Keywords: spider mites should be removed since it appears in the Title.
Introduction: Full names of species should be given, the species are mentioned for the first time. In many places the sentence starts with the abbreviated generic name that is not correct and should be changed.
Line 63, the Authors write about nine species, but I could count only eight listed.
Line 84, remove ‘using the five spider mite species‘, since it is repeated in line 87.
It would be good to present hypotheses tested in this study.
Methods:
Line 94, please write latin name with Italic font.
Line 95, not clear why it is writen ‘in a climate room in the laborator‘. Only this species?
Line 103, not clear what do you mean by ‘spider mites are identified through the nuclear ribosomal transcribed spacer (ITS) region‘.
Line 102 and other lines: expression ‘cage‘ sounds a bit strange to me, unlesss it is used in spider mites terminology.
Line 113, unclear ‘plants with the same growth‘
Line 113, is the any reason why the weight was 2.35 not e.g. 2 g?
You do not refer to Figure 1. However I am not sure if this figure is needed. Was this methd somehow different that used in other references? Could not you just refer tot he reference that described this method first?
Lines 142 and 152, sentence can not start with a symbol.
Results: No reference to Fig. 2 in the text. Title of Fig. 2 – abbreviations of species names should be unified.
Line 174, please add space before (G)
Line 175, please add space before NC.
Liners 192-195, it fits better to Discussion not to Results..
References: Please unify the system of writing doi and remove underlined fonts (e.g., lines 286-289).
Author Response
Thank you for your suggestions. We have seriously studied the comments and made a major revision accordingly. The attached file is a point-by-point response to each of the comments and suggestions. We are grateful for your helpful comments which improved the manuscript a lot.

Reviewer 2 Report
Dear Authors,
Thank you for the invitation to meet your manuscript titled:
"Phylogenetic related divergence in perceiving suitable host 2 plants among five spider mites species"
The manuscript is generally well written.The results are presented
clearly. Appropriate methods of analysis have been used. I have
only minor comments:
Line 41 and Line 44 - Ordered for the first time the full species name i.e. Tetranychus urticae
Tetranychus kanzawai.......
- Line 46 and 47 - Ordered for the first time the full species name i.e. Tetranychus
According to the rules, the first time is always given the full species name.
- No citations in content, in text - for Table 1, Figure 1,
- 2.5. Data Analysis. At what level of probability alpha the analyses were performed? Provide a notation
- - Line 153 - Fig. 3 should it be fig 2
Author Response
Thank you for your suggestions. We have revised our manuscript accordingly. The attached file is a point-by-point response to each of the comments and suggestions.

Reviewer 3 Report
Dear Colleague,
The paper is interesting and well written. I just made few comments and suggestions in the attached file.
In my opinion, the paper can be published after these suggestions and corrections.
Best regards.

Author Response
Response: Thank you for your suggestions. We have fixed all the six points as you raised and suggested.
Round 2
Reviewer 1 Report
The Authors considerably improved the MS and in my opinion it can be now accepted for the publication in the journal Insects.
I have only two comments:
About Line 254: it should be added in the Methods which software you used to create ML tree.
Line 265: please delete part of te sentence: 'Our PCAs show that'
Author Response
About Line 254: it should be added in the Methods which software you used to create ML tree.
Response:Sorry, we might not clearly explain about how we got the tree. We just summarized the phylogenetic relationships among the five species based on the results of previous studies, and drew the phylogenetic tree in the Photoshop software. Actually, we didn’t do phylogenetic analyses because we thought the phylogenetic relationships among the five species was very clear. We just used this information in Discussion. So, we think it is not necessary to add this part to Methods.
Line 265: please delete part of te sentence: 'Our PCAs show that'
Response: Done. (line 264)